# Antitumor Activity of DFX117 by Dual Inhibition of c-Met and PI3Kα in Non-Small Cell Lung Cancer

**DOI:** 10.3390/cancers11050627

**Published:** 2019-05-05

**Authors:** Yanhua Fan, Huaiwei Ding, Donghwa Kim, Duc-Hiep Bach, Ji-Young Hong, Yongnan Xu, Sang Kook Lee

**Affiliations:** 1College of Pharmacy, Natural Products Research Institute, Seoul National University, Seoul 08826, Korea; fanyhkyem@163.com (Y.F.); kdh9298@gmail.com (D.K.); bdhiep90@gmail.com (D.-H.B.); jyhong7876@daum.net (J.-Y.H.); 2The Key Laboratory of Chemistry for Natural Products of Guizhou Province, Chinese Academy of Sciences, Guiyang 550014, China; 3Key Laboratory of Structure-Based Drug Design and Discovery, Ministry of Education, Shenyang Pharmaceutical University, Shenyang 110016, China; dinghuaiwei627@163.com

**Keywords:** dual inhibitor of PI3K and Met, 3-substituted imidazo[1,2-a]pyridine (DFX117), cell cycle arrest, apoptosis, non-small cell lung cancer (NSCLC)

## Abstract

Aberrant activation of hepatocyte growth factor (HGF)/c-Met signaling pathway caused by gene amplification or mutation plays an important role in tumorigenesis. Therefore, c-Met is considered as an attractive target for cancer therapy and c-Met inhibitors have been developed with great interests. However, cancers treated with c-Met inhibitors inevitably develop resistance commonly caused by the activation of PI3K/Akt signal transduction pathway. Therefore, the combination of c-Met and PI3Kα inhibitors showed synergistic activities, especially, in *c-Met* hyperactivated and *PIK3CA*-mutated cells. In our previous study, we rationally designed and synthesized DFX117(6-(5-(2,4-difluorophenylsulfonamido)-6-methoxypyridin-3-yl)-N-(2-morpholinoethyl) imidazo[1,2-a]pyridine-3-carboxamide) as a novel PI3Kα selective inhibitor. Herein, the antitumor activity and underlying mechanisms of DFX117 against non-small cell lung cancer (NSCLC) cells were evaluated in both in vitro and in vivo animal models. Concurrent targeted c-Met and PI3Kα by DFX117 dose-dependent inhibited the cell growth of H1975 cells (*PIK3CA* mutation and *c-Met* amplification) and A549 cells (KRAS mutation). DFX117 subsequently induced G0/G1 cell cycle arrest and apoptosis. These data highlight the significant potential of DFX117 as a feasible and efficacious agent for the treatment of NSCLC patients.

## 1. Introduction

c-Met receptor tyrosine kinase (RTK), activated by its ligand hepatocyte growth factor (HGF), is frequently hyperactivated, amplified or mutated in many human cancers including lung cancer [1,2,3,4]. The binding of HGF with c-Met leads to c-Met autophosphorylation at tyrosine residues Y1234 and Y1235 within the activation loop of the kinase domain and then eventually phosphorylates at tyrosines Y1349 and Y1356 to activate c-Met-mediated signaling pathway. Sustained activation of c-Met signaling triggers multiple signaling cascades which subsequently regulate various cellular processes including cell survival, proliferation, migration, invasion and apoptosis such as PI3K/Akt, MEK/MAPK and STAT3 pathways [1,2,3,4]. In addition, *c-Met* gene amplification or mutation is also conferred to intrinsic resistance to epidermal growth factor receptor-tyrosine kinase inhibitors (EGFR-TKIs) [5,6,7]. There is an increasing number of evidence that show that the c-Met inhibitor exhibits a strong synergistic effect with EGFR inhibitors on EGFR-TKI resistant non-small cell lung cancer (NSCLC) cells harboring acquired *c-Met* gene amplification [8,9,10]. Therefore, c-Met is considered as an attractive target biomarker for cancer therapy, particularly for EGFR-TKI resistant cancer. In line with this, a diverse class of c-Met inhibitors has been developed as anticancer agents for c-Met-driven tumors [11,12,13].

The continuous use of c-Met inhibitors develops drug resistance which commonly occurs through the activation of Phosphatidylinositol-4,5-bisphosphate 3-kinase (PI3K)/Akt signaling, amplification of *c-Met* and *KRAS* mutation [14,15]. Mutations in members of the PI3K pathway are most commonly encountered in the *PIK3CA*, which encoded p110α catalytic subunit of PI3K, and tumors harboring *PIK3CA* mutations remain active upon c-Met inhibition, which render drug resistance to c-Met inhibitors [16,17]. Thus, it is quite clear that combination of c-Met and PI3Kα inhibitors might have synergistic activity, especially in *c-Met* hyperactivated, EGFR T790M and *PIK3CA*-mutated cell lines (e.g., NCI-H1975 cells). Currently, increased evidence of clinical data shows the limitation of activity of single PI3K inhibitor at tolerated doses [17,18]. However, combined targeting of PI3K and c-Met could improve the tolerability and increase efficacy compared to a single PI3K inhibition or c-MET inhibition [16,17,18]. Supposedly, a dual inhibitor of PI3Kα and c-Met might be an effective therapeutic agent to overcome drug resistance or enhance the efficacy of single treatment.

Somatic *KRAS* mutations also strongly decrease the effectiveness of c-Met inhibitors through sustained ERK, MAPK and PI3K activation [19,20]. It suggests that simultaneously targeting both c-Met and KRAS might be an effective strategy when both oncogenic drivers are overexpressed [20,21]. Therefore, the development of a dual inhibitor of PI3Kα and c-Met could provide therapeutic benefits specifically to patients with *c-Met* amplification and *EGFR* mutation or *KRAS* mutation NSCLCs.

We recently designed and synthesized a new 3-substituted imidazo[1, 2-a]pyridine derivative, named DFX117 (6-(5-(2,4-difluorophenylsulfonamido)-6-Methoxypyridin-3-yl)-N- (2-morpholinoethyl)imidazo[1,2-a]pyridine-3-carboxamide), which exhibited a potent PI3Kα inhibitory activity with IC_50_ value of 0.5 nM [22]. The present study revealed that DFX117 is also a potent c-Met tyrosine kinase inhibitor. Importantly, DFX117 exhibited a favorable antitumor activity against NSCLC cells harboring *c-Met* amplification, *EGFR* and *KRAS* mutation. Herein, we report studies on the antitumor activity and the underlying mechanism of DFX117 against NSCLC cells NCI-H1975 (*c-Met*-amplified, PI3K-mutated, and *EGFR*-mutated cells) and A549 (*KRAS* mutated cells).

## 2. Results

### 2.1. DFX117 Exhibits Anti-Proliferative Activity of Lung Cancer Cells

Our previous study revealed that DFX117 is a selective PI3Kα inhibitor with an IC_50_ value of 0.5 nM in cell-free assays [22]. DFX117 also exhibited the growth inhibitory activity against various cancer cells including the A549 cells [22]. Considering the role of the PI3K/Akt signaling pathway in lung cancer development, we further extended to evaluate the anti-proliferative activity of DFX117 in cultured several human lung cancer cell lines (NCI-H1975, NCI-H1993, and HCC827). DFX117 significantly inhibited the growth of all tested lung cancer cell lines with IC_50_ values ranging from 0.02 to 0.08 μM (Figure 1A,C). Among the tested cell lines, the NCI-H1975 cells were the most sensitive to DFX117 with an IC_50_ value of 0.02 μΜ. Therefore, further analysis of DFX117 to elucidate the plausible mechanisms of action in the antitumor activity was performed in the A549 (wild-type *PIK3CA* and *EGFR*) and NCI-H1975 (mutated *PIK3CA* and *EGFR*) cells [23]. The growth inhibitory activity of DFX117 in both A549 and NCI-H1975 cells was found to be superior to that of HS-173, a known PI3Kα inhibitor [21]. DFX117 showed selective growth inhibition of lung cancer cells compared to normal cell lines MRC-5 (human fetal lung fibroblasts) with the IC_50_ values of approximately 40 times difference (Figure 1B,C). The colony formation was also significantly inhibited by DFX117 (Figure 1D,E), which is consistent with the result of the growth-inhibitory activity performed by the SRB assay (Figure 1A).

### 2.2. DFX117 Suppresses the PI3K/Akt/mTOR Signaling Pathways in Lung Cancer Cells

To further elucidate the anticancer mechanism of DFX117, the regulation of PI3K signal transduction pathway associated with cancer cell growth was analyzed using Western blot analysis. After DFX117 treatment for 24 h, the protein levels of PI3K signaling pathways including p-Akt, p-mTOR, p-p70S6K, p-GSK3β, p-4EBP1 and p-eIF4E were effectively suppressed in both A549 and NCI-H1975 cells (Figure 2A,B). In contrast, the expression of PTEN, a tumor suppressor was enhanced by the treatment of DFX117 in both cells (Figure 2A,B). The suppressive effect of DFX117 on p-Akt expression was also manifested by observation of immunofluorescence analysis under a confocal microscope after treated with DFX117 (0.2 µM) for 8 h in A549 (Figure 2C) and NCI-H1975 cells (Figure 2D,E). Interestingly, DFX117 effectively suppressed the expression of mRNA of *PIK3CA* in a concentration-dependent manner, which is different from other PI3K kinase inhibitors (Figure 2F,G).

### 2.3. DFX117 Inhibits c-Met Kinase Activity

To further determine the possible target of DFX117, we employed phospho-RTK array analysis using A549 cells. Data suggested that up-regulated phospho-Met level was markedly suppressed upon DFX117 (0.2 μM) treatment (Figure 3A). Based on these data, we assumed that the Met might be one of plausible molecular targets for DFX117-mediated growth inhibition of cancer cells. Therefore, we further evaluated the effects of DFX117 on Met activity employing molecular docking study and cell-free kinase activity assays. As shown in Figure 3B, DFX117 formed a single typical hydrogen bond between the nitrogen atom of imidazo[1,2-a]pyridine moiety and the back-bone carbonyl of Met1160, which is a characteristic of c-Met inhibitors bound to ATP binding site in kinase domains. The sulfonamide group forms another two hydrogen bonds with residues Asp1164 and Asn1167. Moreover, the morpholinyl formed an additional hydrogen bond with the back-bone NH of Asp1222. All these hydrogen bonds would stabilize the conformations and interactions between DFX117 and ATP binding site of c-Met in kinase domains with lower energy. DFX117 formed hydrophobic interactions with c-Met to fit the hydrophobic pocket composed by Ile1084, Met1211, Ala1108, Tyr1159 and Gly1163. DFX117 also effectively inhibited the cell-free kinase activity of c-Met with an IC_50_ of 17.1 nM (Figure 3C). In the same experimental condition, crizotinib, a positive control, exhibited an IC_50_ value of 11 nM. These data suggest that DFX117 is a potent Met kinase inhibitor and the down-regulation of phospho-Met level might be associated with the inhibition of Met kinase activity in the cells.

### 2.4. DFX117 Suppresses the Met Signaling Pathway in Lung Cancer Cells

Since DFX117 exhibited Met kinase inhibition activity, further study was conducted to assess the effect of DFX117 on Met signaling pathways. Primarily, the levels of activated Met (phospho-Met) and Met-mediated downstream molecules were determined by Western blot analysis after treatment of DFX117 for 24 h in A549 and NCI-H1975 cells. As shown in Figure 3D,E, the levels of phosphorylated Met, activation forms of Met, were significantly suppressed by DFX117 in a concentration-dependent manner in both cells. The suppression of Met activation by DFX117 subsequently down-regulated the Met-mediated major downstream molecules including p-RAC1, p-STAT3 and p-ERK1/2, which are highly associated with the cell proliferation and survival, in a concentration-dependent manner without obvious alterations of each of total protein levels.

### 2.5. DFX117 Induces G1 Cell Cycle Arrest and Apoptosis in Lung Cancer Cells

It is well-known that the PI3K and Met signaling pathways are highly associated with the regulation of cell cycle checkpoints in cancer cell proliferation. To further evaluate the effects of DFX117 on the cell cycle regulation, the cells were treated with DFX117 for 24 h, and cell cycle distribution was detected by FACS analysis. DFX117 induced cell cycle arrest at the G0/G1 phase in a concentration-dependent manner compared to that of the control groups in both A549 and NCI-H1975 cells (Figure 4A,B). The effects of DFX117 on cell cycle regulation were further confirmed by Western blot analysis. As shown in Figure 4C,D, the phospho-cyclin D1 at Thr286, which enhanced ubiquitination and proteasomal degradation of cyclin D1, was increased by DFX117 with a decrease in cyclin D1. In addition, the expression levels of c-Myc, CDK2/4, cyclin E, cyclin A, Rb and p-Rb were significantly suppressed, but the stabilization of p27 and p21 expression levels was upregulated by DFX117 treatment for 24 h in a concentration-dependent manner (Figure 4C–F). These findings suggest that the anti-proliferative activity of DFX117 in cancer cells is in part correlated with the induction of G0/G1 cell cycle arrest via its regulation of the expression levels of proteins in the PI3K/Akt and c-Met pathways.

To further evaluate whether DFX117 is able to induce apoptotic cell death after an extended exposure time in lung cancer cells, the cells were treated with DFX117 for 48 h, and Annexin V-positive cells were detected by FACS analysis. As shown in Figure 5A,B, the population of apoptotic cells (Annexin V positive) was increased in a concentration-dependent manner in both A549 (from 6.09% to 26.84% at 0.4 μM DFX117) and NCI-H1975 (from 4.15% to 32.75% at 0.4 μM DFX117) cells. These data indicated that the longer exposure of DFX117 induced apoptotic cell death in both lung cancer cell lines. To further clarify the molecular mechanisms involved in the induction of apoptosis by DFX117, the effect of DFX117 on the expression levels of apoptosis-associated proteins was examined by Western blot analysis after treatment with DFX117 for 48 h in A549 and NCI-H1975 cells. In comparison to the vehicle-treated control group, the p53, Bax and Bad expressions were upregulated by DFX117 treatment, and the anti-apoptotic protein Bcl-2 was downregulated (Figure 5C,D). DFX117 also triggered the activation of the caspase-dependent apoptotic cascade. The cleaved caspase-9 and cleaved caspase-3 expressions were remarkably increased when treated with DFX117 at 0.4 μM, and subsequently elevated the expression level of cleaved PARP in both A549 and NCI-H1975 cells (Figure 5C,D). Taken together, these data suggested that DFX117 induced apoptotic cell death via the caspase (intrinsic) pathway in A549 and NCI-H1975 cells.

### 2.6. DFX117 Inhibits Tumor Growth in a Xenograft Mouse Model

The antitumor activity of DFX117 was further determined in a nude mouse xenograft model implanted with A549 (3 × 10^6^ cells/mouse) and NCI-H1975 human lung cancer cells (7 × 10^6^ cells/mouse). When the tumor volume reached approximately 100 mm^3^, DFX117 (5, 10 or 20 mg/kg) was orally administered daily for 23 days in A549 xenograft model. At the termination of the experiment (30 days after inoculation), the tumor volumes in the vehicle (normal saline)-treated control group were approximately 1100 mm^3^. Compared to that of the control group, tumor growth was significantly inhibited by DFX117 treatment. The inhibition rates of the tumor volume relative to the control group were 48.1%, 67.3% and 79.9% at 5 mg/kg, 10 mg/kg and 20 mg/kg DFX117, respectively, in A549 cells-implanted xenograft model (Figure 6A). No overt toxicity or body weight change was observed with DFX117 administration (Figure 6B). The excised tumor weights on the termination day were also significantly inhibited by DFX117 treatment (Figure 6C). Similarly, in NCI-H1975 xenograft model, the inhibition rates of the tumor volume relative to the control group were 52.0% and 81.89% at 10 mg/kg and 20 mg/kg DFX117 treatment, respectively (Figure 6G). No overt toxicity or body weight change was observed, and the tumor weights were found to be significantly inhibited by DFX117 administration in NCI-H1975 cells-implanted nude mouse models (Figure 6H,I). These data indicated that the oral administration of DFX117 exerted a potential antitumor activity without toxicity in animal models. In addition, immunohistochemical analysis of tumor tissues showed that administration of DFX117 effectively suppressed the expressions of p-Met and p-Akt (Figure 6D,J). The expression level of cleaved caspase-3 was remarkably increased, and effectively suppressed the expression of Ki67 which is a cell proliferation marker, in the DFX117 treatment group compared to vehicle-treated control groups (Figure 6D). We also confirmed the effects of DFX117 on mRNA levels of PIK3CA in tumors. In accordance with the in vitro results, mRNA levels of PIK3CA were dose-dependently decreased in DFX117-treated tumors (Appendix A). In addition, the protein expression levels of p-Met, Grb2, p-GSK3β, p-mTOR, p-STAT3 and p-ERK1/2 were also downregulated by DFX117, suggesting that the Met and PI3K/Akt signaling pathway was inhibited by DFX117 in the in vivo model (Figure 6E,F). The change in the expression levels of proteins related to apoptotic cell death, including p53, Bax, cleaved caspase-9, caspase-3, PARP and Bcl-2, in the tumor tissues was also consistent with the findings from the in vitro cell culture system (Figure 6E,F). Overall, these data suggested that DFX117 effectively inhibited the tumor growth of lung cancer cells in vivo and that the antitumor activity of DFX117 might be partly associated with the downstream regulation of the Met/PI3K/Akt pathway. As shown in Figure 6K,L, the protein expression levels of p-Met as well as its downstream components such as Grb2, p-GSK3β, p-mTOR, p-STAT3 and p-4ERK1/2 were also downregulated by DFX117, suggesting that the Met and PI3K/Akt signaling pathway were in part involved in the antitumor activity of DFX117 in the NCI-H1975 cells-implanted xenograft model. 

### 2.7. Pharmacokinetic (PK) Study of DFX117

Since DFX117 exerted a promising profile for antitumor activity in in vitro cell culture and in vivo xenograft mouse model as a dual PI3Kα and c-Met inhibitor, we further investigated its pharmacokinetic parameters in male ICR mice. The oral administration of 10 mg/kg DFX117 exhibited that the mean residence time (MRT) was 2.2 h and AUC was 3340.6 h·ng/mL (Table 1). When administered orally, DFX117 was quickly absorbed with a Tmax of 0.3 h. The overall PK parameters of DFX117 seem to be promising and encouraging its further evaluation for development of antitumor agents.

## 3. Discussion

Numerous tyrosine kinase inhibitors (TKIs) targeting PI3K and c-Met has been approved (Idelalisib and Crizotinib) or are tested in clinical trials [4,7]. Although TKIs are initially very promising, their efficacy is limited caused by the development of drug resistance [16,17,18,24,25]. Recent studies indicated that PI3Kα was overexpressed or mutated in Met inhibitor resistant cells and tumor xenografts, suggesting co-targeting PI3Kα might be highly promising anticancer therapeutics for Met inhibitor resistant cancer [16,17,18,24,25]. The A549 cells with mutant *KRAS* and wild *EGFR* are rarely respond to the therapy of EGFR-TKI and Met inhibitor, indicating that the benefits of EGFR-TKI or Met inhibitor therapy can be impaired by *KRAS* mutation [26,27,28,29,30]. The NCI-H1975 cells harboring Met amplification, EGFR (T790M and L858R) mutation and *PIK3CA* mutation which contribute to EGFR-TKI resistance showed very low chemosensitivity to EGFR-TKI in preclinical trial [31,32,33]. Therefore, there remains an urgent need for effective and safe therapeutic agents for the treatment of these kinds of TKI resistance tumors and combination therapies can be an effective long-term treatment of cancer with TKI resistance [10,34,35].

In this study, we found that DFX117, designed and synthesized as a PI3Kα inhibitors in the previous study, showed remarkable antitumor activity in both A549 and NCI-H1975 in vitro and in vivo, particularly in NCI-H1975 cells with mutations of EGFR and *PIK3CA* as well as *c-Met* amplification. The data suggested that DFX117 might be a potential candidate agent for all these mutant tumors in RTK signaling pathway. This was also further confirmed by the significant inhibitory effect on cell proliferation of HCC827 cells (*EGFR* mutation) and H1993 cells (*c-Met* amplification). In addition, compared to a highly selective PI3Kα inhibitor HS-173, DFX117 exhibited the more potent growth inhibition of these tested cancer cells in vitro as well as tumor growth in in vivo xenograft model. The data suggested that DFX117, the rationally designed a dual inhibitor of PI3Kα and Met, improved physicochemical properties and enhanced antitumor activities.

It is known that afatinib resistant NCI-H1975 tumor clones showed lower PTEN expression levels than control clones, suggesting the decreased PTEN expression might contribute to the resistance to EGFR-TKI therapy [32,35]. In addition, PTEN restoration attenuated the tumor growth and development induced by Met activation or amplification [35,36]. PTEN affects c-Met dependent signaling at multiple levels including counteracting the effects of PI3K on Akt, regulation of c-Met-dependent gene expression and dephosphorylate proteins that are activated by c-Met [32,37,38]. Therefore, it is generally considered that combining with c-Met inhibitor and PTEN restoration has a beneficial anticancer effect. The present studies showed that DFX117 effectively enhanced the PTEN expression in A549 and NCI-H1975 cells. The up-regulation of PTEN expression by DFX117 might be one of reasons in the enhancement of the efficacy in the anti-proliferation and apoptosis induced by EGFR-TKI in NSCLC [37,38,39].

Met amplification is a well-established paradigm ligand-independent Met activation in NSCLC, and impaired Met receptor degradation seems to be an equally important mechanism for sustained aberrant Met signaling [36]. DFX117 significantly down-regulated Met receptor expression in the NCI-H1975 cells and inhibited Met kinase activity. The finding suggests that DFX117 is able to modulate the aberrant activated Met signaling in the *PIK3CA* mutation and *c-Met* amplification NSCLC cells.

In summary, the present study demonstrates a novel compound DFX117, a dual inhibitor of PI3K and Met, with a potential antitumor activity in human lung cancer cells. In particular, DFX117 might be applicable candidate for further prioritization in the development anticancer agents for non-small lung cancer cells with *KRAS* mutation, *EGFR* mutation and *Met* amplification cells.

## 4. Materials and Methods

### 4.1. Cell Lines

Human embryonic kidney cell HEK293, human non-small-cell lung cancer NCI-H1993, and NCI-H1975 cells were purchased from the American Type Culture Collection (Manassas, VA, USA). Human normal lung fibroblast MRC-5 cells and human non-small-cell lung cancer A549 cells were purchased from the Korean Cell Line Bank (Seoul, Korea). Human non-small-cell lung cancer PC-9 cells were provided by Drs. Jin Kyung Rho and Jae Cheol Lee (Asan Medical Center, Seoul, Korea). The A549, PC9, NCI-H1993, and NCI-H1975 cells were cultured in RPMI 1640 medium supplemented with 10% fetal bovine serum (10% FBS, Gibco, Grand Island, NY, USA) and antibiotics-antimycotics (100 units/mL penicillin G sodium, 100 μg/mL streptomycin, and 250 ng/mL amphotericin B). MRC-5 and HEK293 cells were cultured in DMEM medium supplemented with 10% FBS and antibiotics-antimycotics. Cells were incubated at 37 °C with 5% CO_2_ in a humidified atmosphere.

### 4.2. Reagents

Phospho-Met (Tyr1349), phospho-Met (Tyr1003), phospho-Rac1/cdc42(Ser71), Rac1/2/3, phospho-p44/42 MAPK (Erk1/2), phospho-STAT3 (Tyr705), STAT3, Ki-67, p44/42 MAPK (Erk1/2), phospho-PDK1 (Ser241), PDK1, PTEN, PI3 kinase p110α antibody, phospho-PI3 kinase p85 (Tyr458)/p55 (Tyr199), phospho-Akt (Ser473), phospho-Akt (Thr308), phospho-mTOR (Ser2448), mTOR, phospho-GSK-3α/β (Ser21/9), GSK-3β (3D10), phospho-p70S6 kinase (T389), p70S6 kinase, phospho-4E-BP1 (Thr37/46) (236B4), 4E-BP1 (53H11), phospho-eI4FE (Ser209), eI4FE, phospho-Cyclin D1, Cyclin D1, Cyclin E1 (HE12), Cyclin A, PCNA, phospho-Rb (S807/811), Rb, Caspase-3 (8G10), Caspase-9, PARP, Bad, Cleaved caspase-3 (Asp175) (5A1E) and phospho-Met (Tyr1234/1235) antibodies were purchased from Cell Signaling Technology (Danvers, MA, USA). The Met kinase assay/inhibitor screening kit (# KA0055) was purchased from Abnova (Walnut, CA, USA).

### 4.3. Cell Viability Assay

Cell viability assay was performed using SRB assay as described previously [22].

### 4.4. Colony Formation

The cells (500 cells) were seeded in 6-well plates and cultured overnight before treatment with DMSO or DFX117 at indicated concentration for 72 h. The cells were then washed once with PBS and continue to culture with full-growth medium for 10 days, change the culture medium to fresh ones every three days. Cells were stained with 0.1% crystal violet after fixing with 100% methanol.

### 4.5. c-Met Kinase Activity Assay

The effect of DFX117 on Met kinase assay was measured by Met Kinase Assay/Inhibitor Screening Kit (Cat # KA0055, Abnova, Taipei, Taiwan) according to the manufacturer’s instructions.

### 4.6. Western Blotting

Cells were treated with various concentrations of DFX117. Western blot analysis was performed as described previously [22]. Blots were imaged by ImageQuant LAS 4000 (GE Healthcare Life Sciences, Piscataway, NJ, USA).

### 4.7. Immunofluorescence Microscopy

The immunofluorescence staining was carried out as described previously with the indicated antibodies [40]. The confocal dish was coated with 0.2% gelatin before cell culture. After DFX117 treatment for 8 h, cells were washed with PBS and fixed with 4% paraformaldehyde for 20 min. Then, cells were blocked with 1% BSA for 2 h at room temperature and rinsed three times with washing buffer. Cells were incubated with p-Akt (Ser473) at 4 °C overnight before the second antibody incubation for 1 h at room temperature in dark. Finally, the DAPI staining was performed before visualization under confocal microscopy (Leica TCS SP8, Wetzlar, Germany). The images were recorded by a Leica TCS SP8 confocal microscope using a 40× Silicone Oil Objective (40×/1.30 NA).

### 4.8. Cell Cycle Analysis

Cells cycle analysis was carried out as described previously [41]. Cell cycle analysis was analyzed after DFX117 treatment for 24 h. Briefly, cells were plated in culture dishes and cultured with fresh medium without FBS for 12 h. Then, cells were treated with DFX117 for 24 h and remove the supernatant, the treated cells were fixed with 70% ethanol overnight before staining with propidium iodide mixed with RNase. Keep the dying cells under dark conditions at room temperature for 30 min before being subjected to flow cytometry analysis.

### 4.9. Annexin FITC/PI Assay

Annexin FITC/PI assay was performed as described previously [42]. Cell apoptosis was evaluated using Annexin V FITC/PI apoptosis detection kit (BD Biosciences, San Jose, CA, USA) according to the manufacturer’s instruction. Cells were cultured in culture dishes for 12 h and treated with DFX117 for 48 h. Wash the cells with cold PBS buffer before staining with 5 μL of Annexin V-FITC and 5 μL of propidium iodide in the 500 μL 1× binding buffer. Finally, the stained cells were incubated for 15 min at room temperature in the dark before analyzing with FACS Calibur flow cytometer, (Becton Dickinson, San Jose, CA, USA).

### 4.10. Phospho-Receptor Tyrosine Kinase Array

Phosphorylated RTKs were measured with the Human Phospho-RTK Array Kit (ARY001B) from R&D Systems (Minneapolis, MN, USA) according to the manufacturer’s instructions. A total of 250 μg of lysates from the A549 cells treated with DMSO or DFX117 for 8 h were subjected to analysis.

### 4.11. Molecular Docking Studies

Molecular docking was performed according to the previous study [22]. The crystal structure of c-Met (PDB entry code: 2RFN) in complex with AM7 was used for molecular modeling. The AutoDock 4.2 was employed for docking calculations. The protein optimization of c-Met was carried out using Sybyl 7.3, the cocrystallization ligand and water of c-Met was extracted before minimization. Docking parameters and fragmental volumes for the proteins were assigned using the addsol utility in the AutoDock 4.2 program. The size of energy grid box was set to A 60 × 60 × 60 Å with a grid spacing of 0.375 Å based on its cocrystallization ligand. Affinity grid fields were generated using the auxiliary program AutoGrid 4.0. The Lamarckian genetic algorithm (LGA) was used to find the appropriate binding positions, orientations and conformations of the ligands. In each group, the lowest binding energy configuration with the highest % frequency was selected as the group representative. Accelrys Discovery Studio Visualizer 4.5 was used for graphic display.

### 4.12. RNA Extraction and Real-Time PCR

RNA extraction and real-time PCR were carried out as described previously [3]. The following sequences were used: *PIK3CA* forward: CCA CGA CCA TCA TCA GGT GAA C; *PIK3CA* reverse: CCT CAC GGA GGC ATT CTA AAG T.

### 4.13. Pharmacokinetic Studies of DFX117

The male ICR mice (25–30 g, 7–8 weeks of age) were purchased from Koatech Co. (Pyeontak, Kyonggi-do, Korea). Animals were fed with free access to standard chow diet and water. All animal use and care followed the guidelines approved by the Institutional Animal Care and Use Committee of Flow Sciences.INC. (Approval No.: NDDC PK 2018 076 01-DFX117-(4364)). Animals were acclimatized for 1 week at of 23–25 °C temperature, 40–70% relative humidity and 12 h light/12 h dark cycle before study. DFX117 aqueous solution was prepared in a mixture of 10% DMSO, 40% polyethylene glycol (PEG400) in saline and then oral administrated at a dose of 10 mg/kg. Two groups of mice were orally administered either DFX117 (*n* = 5) at 10 mg/kg or vehicle solvent. Approximately 80 µL of blood samples were collected into BD Microtainer plasma separator tubes at times 0.25, 0.5, 1.0, 2.0, 4.0, 6.0 and 8.0 h through the saphenous vein over 24 h post dosing. Blood samples were centrifuged at 15,000 rpm for 5 min and stored in a freezer until analysis. Protein precipitation was conducted on 15 µL of plasma samples using three volumes of acetonitrile containing carbamazepine as an internal analytical standard. After centrifugation, the supernatant was analyzed on a Nexera XR system (Shimadzu, Japan) system coupled with TSQ vantage triple quadruple (Thermo Fisher Scientific, Waltham, MA, USA) system. Chromatographic separation was performed on a Kinetex XB-C18 column (2.1 × 100 mm, 2.6 μm, particle size; Phenomenex, CA, USA) running a linear gradient of solvent A (0.1% formic acid in deionized water) and B (0.1% formic acid in acetonitrile). Detection of the analyte ions was performed in a positive MRM mode, monitoring the following precursor >product ion transitions: m/z 573.1 > 417.8. Analytical data were processed using MassHunter B04.01 (Agilent Korea, Seoul, Korea). Pharmacokinetic parameters were calculated via non-compartmental analysis of the plasma concentration-time profiles using PK Solver.

### 4.14. Tumor Xenograft Studies

All animal experiments were followed the guidelines approved by the Seoul National University Institutional Animal Care and Use Committee (IACUC permission number SNU-170103-2-1). The models of xenograft tumors were set up by subcutaneous injection of 3 × 10^6^ cells (A549) or 7 × 10^6^ cells (NCI-H1975) in 200 μL of PBS. The mice were randomly divided into four groups for A549 (control group, 20 mg/kg, 10 mg/kg or 5 mg/kg of DFX117) and three groups for NCI-H1975 (control group, 20 mg/kg and 10 mg/kg of DFX117). Drug treatment started when their tumor volumes reached to 80–100 mm^3^. Mice of the control group were administered with 0.2 mL vehicle (DMSO: PEG400: saline = 1:4:5) by oral gavage daily and the other group with various doses of DFX117 for 23 days. Body weight and tumor size were measured three times per week. The tumor size was calculated according to the formula V = 0.5 × length × width^2^.

### 4.15. Immunohistochemistry

The immunohistochemistry of the tumors was carried out as described previously with the indicated antibodies [40].

### 4.16. Statistical Analysis

Statistical significance (*p* < 0.05) was assessed using Student’s *t* test or one-way analysis of variance (ANOVA) coupled with Dennett’s *t* test.

## 5. Conclusions

In this study, the antitumor activity and underlying mechanisms of DFX117 against non-small cell lung cancer (NSCLC) cells were evaluated in both in vitro and in vivo animal models. Concurrent targeted c-Met and PI3Kα by DFX117 dose-dependent inhibited the cell growth of H1975 cells (PIK3CA mutation and c-Met amplification) and A549 cells (KRAS mutation). DFX117 subsequently induced G0/G1 cell cycle arrest and apoptosis. The combined observations indicate DFX117 might be applicable candidate for further prioritization in the development anticancer agents for non-small lung cancer cells with KRAS mutation, EGFR mutation and Met amplification cells. In addition, these data highlight the significant potential of DFX117 as a feasible and efficacious agent for the treatment of NSCLC patients.

## Figures and Tables

**Figure 1 cancers-11-00627-f001:**
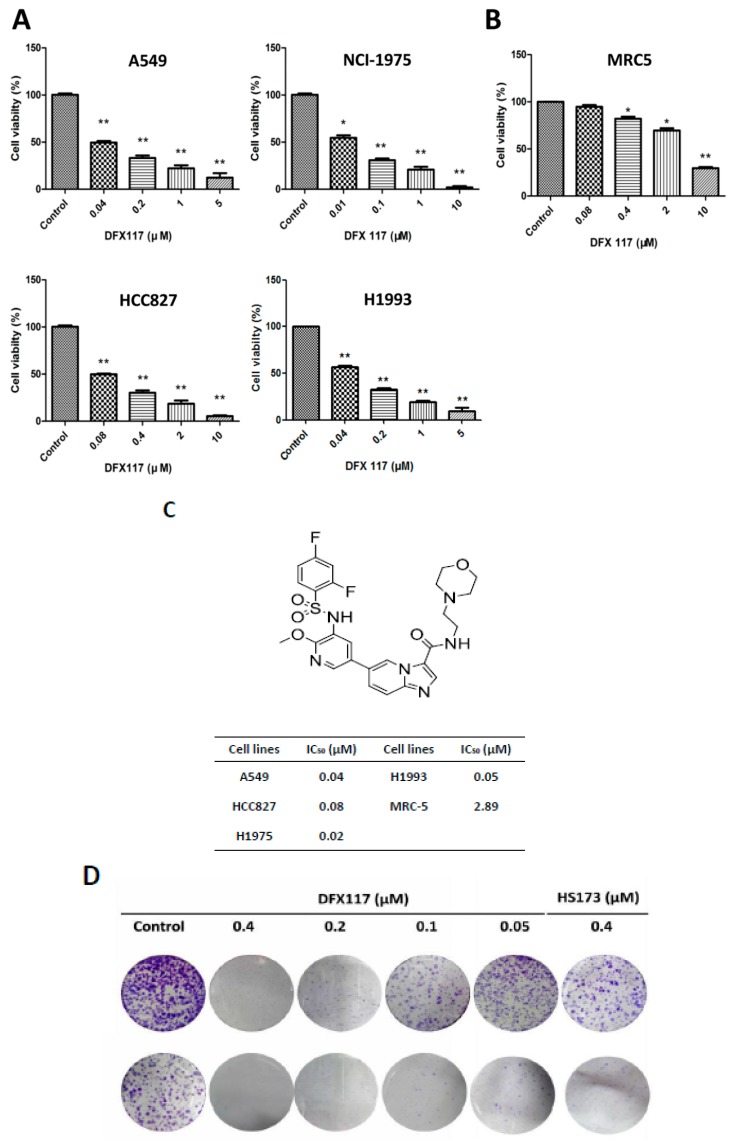
DFX117 significantly induces growth inhibition of non-small cell lung cancer cells. (**A**) The proliferation of four lung cancer cell lines (A549, NCI-H1975, PC9 and NCI-H1993) were significantly inhibited by DFX117. (**B**) DFX117 showed few effects on cell growth of normal cell lines MRC-5 at higher concentration. (**C**) The chemical structure of DFX117 and the IC_50_ values of DFX117 against non-small cell lung cancer cells or normal cell lines. (**D**) Colony formation assays were performed in the A549 and NCI-H1975 cells treated with various concentrations of DFX117. 500 cells were plated in 6 wells plates and incubated for 12 h before treatment with DMSO or DFX117 at indicated concentration for 72 h. Then cells were then washed once with PBS and continue to culture with full-growth medium for 10 days, change the culture medium to fresh ones every three days. Cells were stained with 0.1% crystal violet after fixing with 100% methanol. (**E**) Representative images of the colony formation assay are depicted and compared with HS-173, a PI3Kα inhibitor. * *p* < 0.05 or ** *p* < 0.01 was considered statistically significant compared with the corresponding control values.

**Figure 2 cancers-11-00627-f002:**
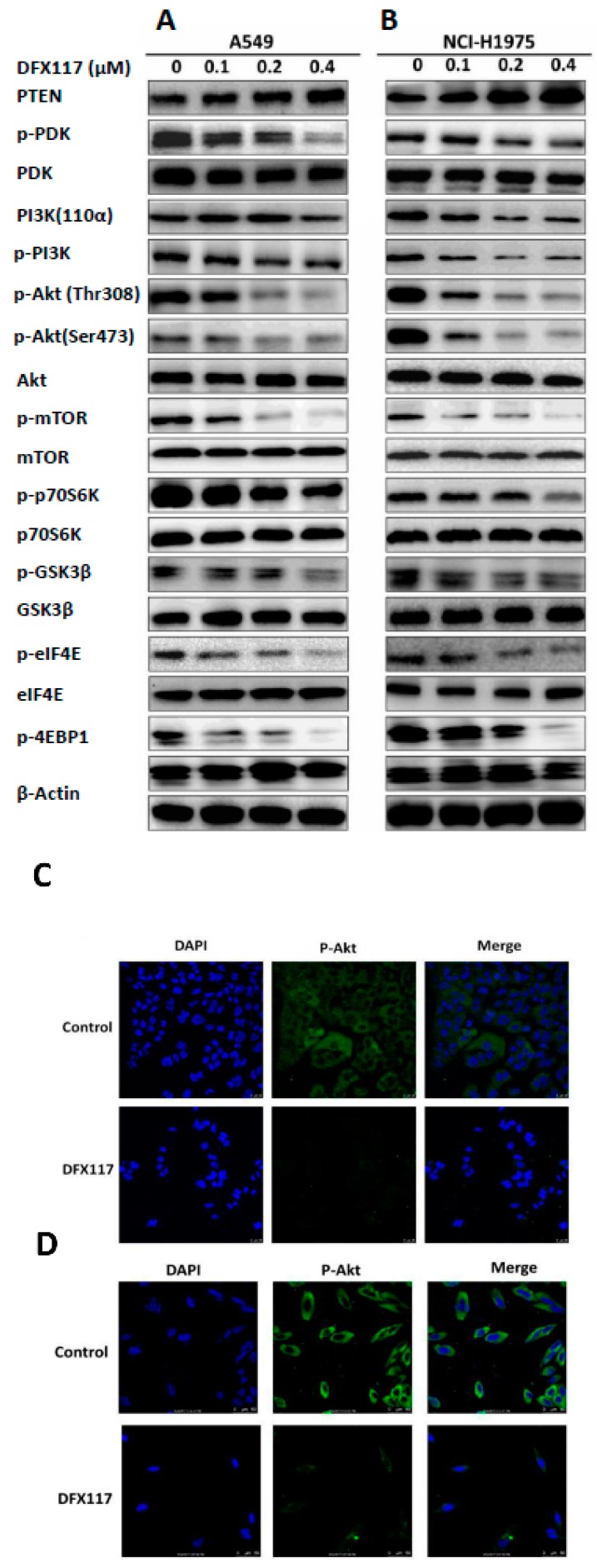
DFX117 suppresses the PI3K signaling pathway. (**A**,**B**) DFX117 suppressed the PI3K- signaling pathway in A549 and NCI-H1975 cells. Cells were collected for Western blot analysis after DFX117 treatment for 24 h at the indicated concentrations. The alterations in PI3K signaling related protein levels were quantified using Image J software. (**C**,**D****)** Immunofluorescence analysis was performed to determine the effects of DFX117 on the expression level of p-Akt (Ser473) in A549 and NCI-H1975 cells. The cells were treated with DFX117 (0.2 μM) for 8 h, and then p-Akt expression was observed under a fluorescence microscope (magnification: 400×). DAPI staining was used to visualize and characterize the nucleus (blue). (**E**) Quantification of the fold changes in the expression levels of p-Akt is compared to the vehicle-treated control (lower panel). (**F**,**G**) DFX117 decreased the mRNA levels of PIK3CA in A549 and NCI-H1975 cells. Cells were treated with DFX117 for 24 h and then collected for RT-PCR analysis. * *p* < 0.05 or ** *p* < 0.01 was considered statistically significant compared with the corresponding control values.

**Figure 3 cancers-11-00627-f003:**
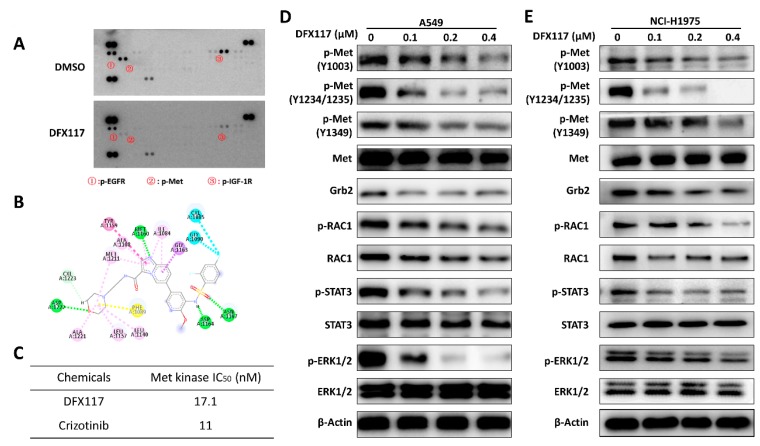
DFX117 inhibit Met kinase activity. (**A**) Phospho- receptor tyrosine kinase (RTK) antibody array after DFX117 treatment. Phosphorylated RTKs were measured with the Human Phospho-RTK Array Kit (ARY001B) from R&D Systems according to the manufacturer’s instructions. A total of 250 μg of lysates from the A549 cells treated with 0.2 μM DFX117 were subjected to analysis. (**B**) Molecular docking model of DFX117 and Met. (**C**) Met kinase activity assay. The effect of DFX117 on Met kinase assay was measured by Met Kinase Assay/Inhibitor Screening Kit according to the manufacturer’s instructions (# KA0055, Abnova, Taipei, Taiwan). (**D**,**E**) DFX117 suppressed hepatocyte growth factor (HGF)/Met signaling pathway. Cells were treated with the indicated concentrations of DFX117 for 24 h.

**Figure 4 cancers-11-00627-f004:**
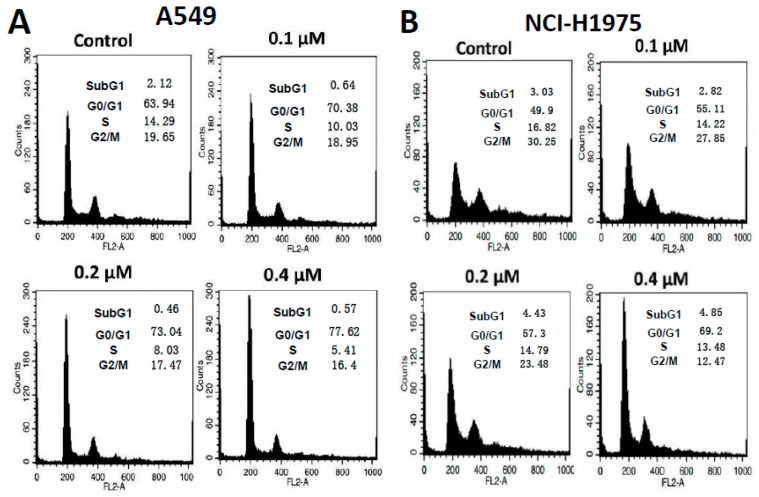
DFX117 altered cell cycle distribution in lung cancer cells. (**A**,**B**) A549 and NCI-H1975 cells were exposed or not exposed to DFX117 at indicated concentrations for 24 h and then were collected for PI (propidium iodide) staining and Western blot analysis. The DNA content was then analyzed by flow cytometric analysis as described in the Materials and Methods. (**C**,**D**) Cells were treated with the indicated concentrations of DFX117 for 24 h, and protein lysates were analyzed
by Western blot to detect the alteration of cell cycle related proteins. (**E**,**F**) The changes in corresponding protein expression levels were quantified using ImageJ. Each bar represents the mean ± SEM (*n* = 3); * *p* < 0.05 or ** *p* < 0.01 was considered statistically significant compared with the corresponding control values.

**Figure 5 cancers-11-00627-f005:**
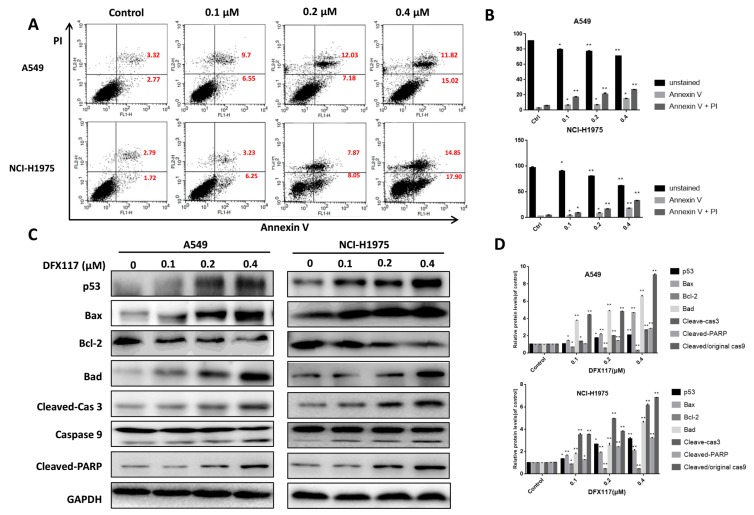
DFX117 induced apoptosis in lung cancer cells. (**A**) Cell apoptosis was analyzed by flow cytometric analysis after Annexin V-FITC/PI staining. Cells were collected and centrifuged at 2000 rpm for 10 min after DFX117 treatment at various concentrations for 48 h. (**B**) The number in the right quadrant of each panel represents the percentage of Annexin V-positive cells. (**C**,**D**) The effect of DFX117 on the induction of apoptosis was also analyzed by Western blot in A549 and NCI-H1975 cells. The changes in corresponding protein expression levels were quantified using Image J. Each bar represents the mean ± SEM (*n* = 3). * *p* < 0.05 or ** *p* < 0.01 was considered statistically significant compared with the corresponding control values.

**Figure 6 cancers-11-00627-f006:**
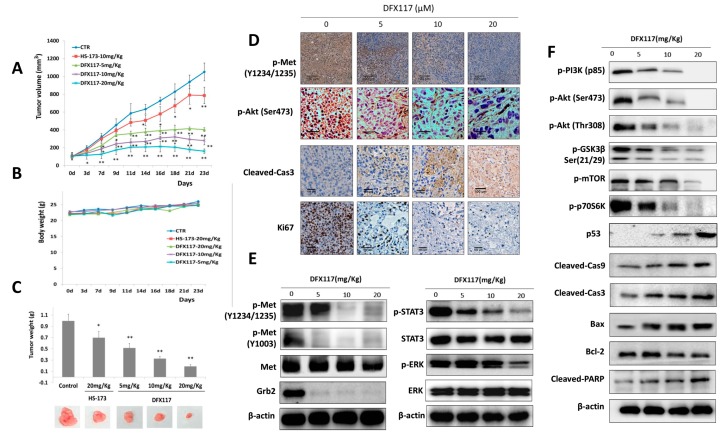
DFX117 inhibited tumor growth in a nude mouse xenograft model. (**A**–**C**) 3 × 10^6^ cells (A549) were subcutaneously injected into nude mice. Mice were orally administered DFX117 (5 mg/kg, 10 mg/kg and 20 mg/kg) or the vehicle control (DMSO: polyethylene glycol: saline = 1:5:4) daily for 23 days after tumor volumes reaching to 100 mm^3^ around. Tumor volumes were measured with calipers and body weights were monitored every 2–3 days. Data represent the means ± SEM (*n* = 5). (**D**) Determination of the alteration of p-Akt (Ser473), p-Met, Ki67 and cleaved caspase 3 in tumor tissues (A549 xenograft). The changes of the related proteins were evaluated by immunohistochemical analysis. Each tissue section was incubated with the indicated antibodies and analyzed by immunohistochemical method described in the Materials and Methods. (**E**,**F**) The effects of DFX117 on the expression levels of the corresponding proteins in Met/PI3K signaling in tumor tissues (A549 xenograft) were examined by Western blot analysis. (**G**–**I**) Mice bearing subcutaneously injected 7 × 10^6^ cells (NCI-H1975) were orally administered DFX117 (10 mg/kg or 20 mg/kg) or the vehicle control (DMSO: polyethylene glycol: saline = 1:5:4) daily for 23 days. Tumor volumes were measured with calipers, and body weights were monitored every 2–3 days. Data represent the means ± SEM (*n* = 5). (**J**) Immunohistochemical analysis of the expression level of p-Akt, p-Met in tumor tissues (NCI-H1975 xenograft). Each tissue section was incubated with the indicated antibodies and analyzed by immunohistochemical method described in the Materials and Methods. (**K**,**L**) The inhibition of DFX117 on Met/PI3K signaling in tumor tissues were analyzed by western blotting (NCI-H1975 xenograft). Data are representative of three independent experiments. Each bar represents the mean ± SEM (*n* = 3). * *p* < 0.05 or ** *p* < 0.01 was considered statistically significant compared with the corresponding control values.

**Table 1 cancers-11-00627-t001:** Pharmacokinetics Parameters of DFX117 after po administration ^a^.

Parameters	T_1/2_ (h)	T_max_ (h)	C_max_ (ng/mL)	AUCall (h·ng/mL)	MRTinf_obs (h)
Value (po ^b^)	2.1	0.3	4504.3	3340.6	2.2

^a^ Values are the average of three runs. Vehicle: DMSO: PEG: saline = 1:4:5. ^b^ Dose: po at 10 mg/mL. T_1/2_, half-life; C_max_, maximum concentration; T_max_, time of maximum concentration; AUCall (h·ng/mL), area under the plasma concentration time curve.

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
