# Peer review of "Antitumor Activity of DFX117 by Dual Inhibition of c-Met and PI3Kα in Non-Small Cell Lung Cancer"

_cancers, 2019, doi:10.3390/cancers11050627_

Reviewer 1 Report

The authors present rigorous and convincing evidence for the potential antitumor activity of DFX117 and data in support of this compound actin as a dual inhibitor of c-Met and PI3K both in cell cullture and in tumor bearing animals. Some strengths of the manuscript are the comparative studies in vitro and in vivo both in terms of tumor growth inhibition and effects on relevant signaling pathways. An additional strength is something that many investigators overlook in studies of antitumor drugs, which is comparison of the therapeutic window between the tumor and normal cells (Figure 1).

Just a couple of minor (but necessary) issues for the authors to address. One is that the extent of apoptosis shown in Figure 5 is not consistent with the insignificant apoptosis presented in Figure 4 ( sub G1 cell population). Another observation that the authors appeared to overlook is the increase in the phosphorylated form of cyclin D1 in the last section of Figure 4. What is the significance of this change in terms of the growth suppression by the drug?

Minor comment. For Figure 4B, the authors might want to omit the unstained bars and just present Annexin V and Annexin V + PI staining with a Y  axis maximum of ~ 40%. This would show the data more clearly.

Author Response

We would like to thank the reviewers for carefully reading our manuscript as well as for the insightful and productive suggestions. Accordingly, we have revised the manuscript and organized the figures as suggested. Below, we describe the results of our overall manuscript revision and responses to the reviewers’ comments as well as to that of the associate editor. The modifications have been marked in red in the manuscript. We hope that the reviewers find our responses clear and to the point.

Reviewer #1:

(1)   The extent of apoptosis shown in Figure 5 is not consistent with the insignificant apoptosis presented in Figure 4 ( sub G1 cell population).

Response: Thank you very much for your careful review and this comment. For cell cycle analysis in Figure 4, cells were treated with DFX117 just for 24 h. In addition, we remove the supernatant before the cell collection. We do did cell cycle analysis for 48 h, however, cell cycle arrest at 48 h was not as significant as that at 24h, it might because of the increased apoptotic cells. So 24 h was finally chose for cell cycle test and 48 h for cell apoptosis analysis. This leads to the different population of apoptotic cells in Figure 4 and Figure 5.

(2)   What is the significance of the increase in the phosphorylated form of cyclin D1 in terms of the growth suppression by the drug?

Response: We thank the reviewer for this comment. The phosphorylated at Thr286 of cyclin D1 will enhance its ubiquitination and proteasomal degradation and then cause cell cycle arrest. That’s why we performed both cyclin D1 and phosphor-cyclin D1 to study the cell cyle effects of DFX117. We have made the corresponding changes according to the Reviewer’s suggestion in the manuscript and marked in red.

(3)   For Figure 4B, the authors might want to omit the unstained bars and just present Annexin V and Annexin V + PI staining with a Y axis maximum of ~ 40%. This would show the data more clearly.

Response: We have made the corresponding changes according to the Reviewer’s suggestion in Figure 4.

With many thanks for your help and consideration.

Reviewer 2 Report

In the manuscript entitled “Antitumor Activity of DFX117 by Dual Inhibition of c-Met and PI3Kα in Non-Small Cell Lung Cancer”, the authors investigated the efficacy and underlying mechanisms of DFX117 against NSCLC. Overall, this is a well-written manuscript with well-executed experimental design and interpretation. There are a few points, which, if addressed, could add value.

1, On page 3 of 19, in the text from row 89-92, the figures cited are not consistent with the data in figure 1.

2, HEK293 is not a lung cell line, the authors might need to avoid using it as a drug selectivity control in figure 1.

3, Row 114, DFX 117 was used at 0.2 uM rather than 0.2 M? Also, similar mistakes were found in row 197?

4, In figures 4 and 5, p21, bad, and bax were upregulated upon DFX 117 treatment; also, the cellular apoptotic pathway was activated. It seems that p53 plays a role in it. To make the study more convincing and detailed, the author might want to refer to and cite this paper “Iron Metabolism Regulates p53 Signaling through Direct Heme-p53 Interaction and Modulation of p53 Localization, Stability, and Function” as well as to detect p53 protein to further find out the p53’s role involved in DFX 117 treatment in vitro and in vivo (mice expt. in fig 6).  

5, In most of the figures, b-actin was over-exposed and the blots should be replaced.

Author Response

We would like to thank the reviewers for carefully reading our manuscript as well as for the insightful and productive suggestions. Accordingly, we have revised the manuscript and organized the figures as suggested. Below, we describe the results of our overall manuscript revision and responses to the reviewers’ comments as well as to that of the associate editor. The modifications have been marked in red in the manuscript. We hope that the reviewers find our responses clear and to the point.

Reviewer #2:

(1) On page 3 of 19, in the text from row 89-92, the figures cited are not consistent with the data in figure 1.

Response: Thank you very much for your careful review. We have checked the cited figures again and revised it.

(2) HEK293 is not a lung cell line, the authors might need to avoid using it as a drug selectivity control in figure 1.

Response: Thank you very much for your careful review. We have revised this part and delete the data about HEK293 in the manuscript and Figure 1.

(3) Row 114, DFX 117 was used at 0.2 uM rather than 0.2 M? Also, similar mistakes were found in row 197?

Response: Thank you very much for your careful review. We have made the corresponding changes according to the Reviewer’s suggestion.

(4) In figures 4 and 5, p21, bad, and bax were upregulated upon DFX 117 treatment; also, the cellular apoptotic pathway was activated. It seems that p53 plays a role in it.

Response: Thank you very much for your careful review and this comment. We have added the data of the effects of DFX117 on p53 in Figure 5 and Figure 6 according to the Reviewer’s suggestion.

(5) Change the over-exposed blots of Actin

Response: We thank the reviewer for this comment. We have replaced some over-exposed blots in the figures.

With many thanks for your help and consideration.

Round  2

Reviewer 2 Report

All my concerns are addressed in the revised version.